# Multiple resistance factors collectively promote inoculum-dependent dynamic survival during antimicrobial peptide exposure in *Enterobacter cloacae*

**Andrew N. Murtha**[1,2], **Misha I. Kazi**[1], **Eileen Y. Kim**[1], **Facundo V. Torres**[1,2], **Kelly M. Rosch**[1], **Tobias Dörr**[1,2,3] *

1 Weill Institute for Cell and Molecular Biology, Cornell University, Ithaca, New York, United States of America, 2 Department of Microbiology, Cornell University, Ithaca, New York, United States of America, 3 Cornell Institute of Host-Microbe Interactions and Disease, Cornell University, Ithaca, New York, United States of America

☯ These authors contributed equally to this work.
* tdoerr@cornell.edu

**Data Availability Statement:** All relevant data are within the manuscript and its Supporting Information files.

## Abstract

Antimicrobial peptides (AMPs) are a promising tool with which to fight rising antibiotic resistance. However, pathogenic bacteria are equipped with several AMP defense mechanisms, whose contributions to AMP resistance are often poorly defined. Here, we evaluate the genetic determinants of resistance to an insect AMP, cecropin B, in the opportunistic pathogen *Enterobacter cloacae*. Single-cell analysis of *E. cloacae's* response to cecropin revealed marked heterogeneity in cell survival, phenotypically reminiscent of heteroresistance (the ability of a subpopulation to grow in the presence of supra-MIC concentration of antimicrobial). The magnitude of this response was highly dependent on initial *E. cloacae* inoculum. We identified 3 genetic factors which collectively contribute to *E. cloacae* resistance in response to the AMP cecropin: The PhoPQ-two-component system, OmpT-mediated proteolytic cleavage of cecropin, and Rcs-mediated membrane stress response. Altogether, our data suggest that multiple, independent mechanisms contribute to AMP resistance in *E. cloacae*.

## Author summary

Antibiotics are losing their efficacy at an alarming rate, necessitating the development of novel strategies to combat bacterial infections. Antimicrobial peptides, which are essentially antibiotics produced by most animals, including humans, have been proposed as an alternative pathway to develop new clinical antimicrobials. However, similar to antibiotics, resistance against AMPs is widespread and varied, yet poorly-understood. Here, we have found that three major factors (the PhoPQ and Rcs cell envelope stress systems, and the protease OmpT) contribute to AMP resistance in the bacterium *Enterobacter cloacae*. We find that resistance is not equally distributed among a population, i.e. some cells are

**Funding:** This research was in part supported by NIH (https://www.nih.gov) grant R01 AI143704 (TD). MIK is supported by a Fleming Postdoctoral Fellowship (https://wicmb.cornell.edu/fleming/). The funders had no role in study design, data collection and analysis, decision to publish, or preparation of the manuscript.

**Competing interests:** The authors have declared that no competing interests exist.

intrinsically more resistant than others. The number of resistant individuals in a population is determined by the three resistance factors we define here. Overall, our work provides new potential targets for the development of antibiotic compounds that may make AMPs work better against dangerous pathogens.

## Introduction

Bacterial pathogens develop resistance to antibiotics at a remarkable rate, complicating our ability to fight life-threatening bacterial infections. This includes the evolution of multi-drug resistant (MDR) infections for which no currently available antibiotic is effective [1]. This is a particular problem with many Gram-negative pathogens, such as prominent Enterobacterales. *Enterobacter cloacae*, for example, is an opportunistic, rod-shaped Gram-negative pathogen, which can cause respiratory, soft tissue, and bloodstream infections [2]. While it is a member of our typical gut microflora, MDR *E. cloacae* have been designated a major public health risk (the genus *Enterobacter* is the second "E" in the infamous ESKAPE high priority pathogen list [3]), particularly in immunocompromised patients [4]. Similar risks exist for other Enterobacterales, such as *Escherichia coli* [5] and *Klebsiella pneumoniae* [5,6]. It is therefore essential to explore and develop alternative antimicrobials capable of eradicating MDR infections.

Antimicrobial peptides (AMPs) are short, often cationic peptides produced by nearly all domains of life, making up a key component of innate immunity [7–10]. This includes humans, which produce over 100 different AMPs to defend epithelial surfaces and to aid phagocytes in bacterial killing. AMPs also comprise a portion of our initial systemic response to infection [11]. Thousands of structurally unique AMPs have been isolated from other (even extinct) mammals, amphibians, reptiles, fish, birds, arachnids (tarantulas, araneomorph spiders, and scorpions), and mollusks [12–15]. Invertebrates lack adaptive immunity and rely heavily on AMPs to ward off invading microbes [16,17].

Most positively-charged AMPs bind to negatively-charged bacterial membranes [7]. Binding may occur in an orderly, structured manner to create pores in the membrane. This may disrupt the permeability barrier, dissipate ion gradients, or promote entry of other AMPs with intracellular targets, which will result in the death of the bacterial cell [18]. In other cases, AMPs will coat the cell surface. This causes a detergent-like effect, which scrambles the integrity of the cell envelope and results in lysis [19]. Importantly, AMPs tend to kill bacteria more quickly than antibiotics [20] with a more narrow range of concentrations eliciting responses from the bacterial cell [21]. In principle, this means fewer opportunities for a population of bacteria to develop resistance mechanisms to AMPs, at least through mutation.

Due to their natural abundance, diversity, and potent antibacterial activity, AMPs are of high interest as potential alternatives to classical antibiotics [22]. One such example of AMPs being explored for therapeutic purposes are cecropins, a family of linear α-helical AMPs isolated from insects [23,24]. Cecropins display potent antimicrobial activity against several species of Gram-positive and Gram-negative bacteria, but predominantly target Gram-negative LPS [25–28]. Cecropin B, for example, is an amphipathic helix-hinge-helix peptide with the sequence KWKVFKKIEKMGRNIRNGIVKAGPAIAVLGEAKAL. It rapidly kills both Gram-negative and Gram-positive bacteria, with no observed resistance development [29]. Cecropin B is more potent against Gram-negative bacteria [25,30], and ineffective against Gram-positive spheroplasts [29], suggesting the Gram-negative outer membrane (mainly composed of lipopolysaccharide, LPS) as the major point of attack, though the exact mechanism of action is unknown. Dozens of other AMPs are currently in clinical development [31]. Therefore, it is

vital to understand the mechanisms by which pathogens can resist killing by AMPs. This may enable the more rational design of AMPs for future therapeutic use.

The main characterized AMP resistance mechanisms are target (membrane) modification, AMP degradation, efflux and exclusion of AMPs. One of the most well-understood mechanisms for AMP resistance is modification of the outer membrane (OM) [32,33]. For example, addition of amino sugars [34], phosphoethanolamine [35], or other amine-containing residues, or removal of phosphate residues [36], prevents the electrostatic binding of AMPs to normally negatively charged lipopolysaccharide (LPS), the primary component of the bacterial OM [37]. The rigidity of the OM likely also impacts AMP susceptibility; several members of Enterobacterales attach additional acyl chains to their lipid A in response to AMP exposure, modifying the packing and fluidity of LPS molecules [38]. In addition to membrane modification, many bacteria produce proteases capable of degrading AMPs. This includes secreted proteases [39], membrane-bound enzymes [40], or even AMP importers for intracellular cleavage [41,42]. The susceptibility of an AMP to proteolytic cleavage is highly dependent on its structure, and linear AMPs may be more prone to degradation than an AMP with multiple disulfide bonds [43]. Additionally, biofilms can provide an additional layer of protection, hampering the ability of AMPs to reach the bacterial cell envelope. Exopolysaccharides and capsular polysaccharides (CPS) have been shown to confer up to 1000-fold higher AMP resistance in capsulated vs non-capsulated strains of many bacteria [44,45]. In Enterobacterales, production of some CPS is regulated by the envelope stress response system known as the Rcs phosphorelay [46]. More specifically, Rcs positively regulates the production of a colanic acid (CA) capsule, which has been implicated in resistance to antibiotics and antimicrobial peptides [47]. Indeed, induction of colanic acid by sub-MIC antimicrobial peptides can "prime" *E. coli* for enhanced survival after a second, higher dose [48].

Complicating the study of these resistance mechanisms is so-called "heteroresistance". In a heteroresistant population of bacteria, a subset of cells within the genetically identical population displays higher resistance to an antibiotic or antimicrobial peptide [49–52]. Distinct from outright resistance, heteroresistance is often not stable and is instead a result of varying temporal levels of gene expression or unstable genetic alterations [53]. As heteroresistance can complicate our ability to diagnose and treat bacterial infections, it is important to investigate the nuanced mechanisms that drive this and other seemingly stochastic phenotypes. Of note, the reason for the stochastic nature of heteroresistance (i.e., the underlying cause of population heterogeneity) is poorly understood.

In this study, we used the opportunistic pathogen *Enterobacter cloacae* and the insect AMP cecropin B (cecB) as a model system to study mechanisms responsible for AMP resistance in Enterobacterales. We observed inoculum-dependent killing of *E. cloacae* by cecB, with cells at high inoculum conditions ultimately displaying higher resistance to killing after an initial die-off, than cells at low inoculum. Single-cell analysis revealed this growth was due to "dynamic survival" within the population (a dynamic state of growth and death), uncovering the underlying cause of the inoculum effect against AMPs. We then identified the PhoPQ two-component system, the OM protease OmpT, and Rcs signaling as collective contributors to dynamic survival.

## Results

### Enterobacter cloacae is resistant to cecB in an inoculum-dependent manner

It has been shown previously that cecropin B (cecB) is effective at killing Gram-negative bacteria [29], but previous experiments have been mostly conducted with well-studied model organisms like *E. coli*. As part of an ongoing investigation into AMP resistance mechanisms in

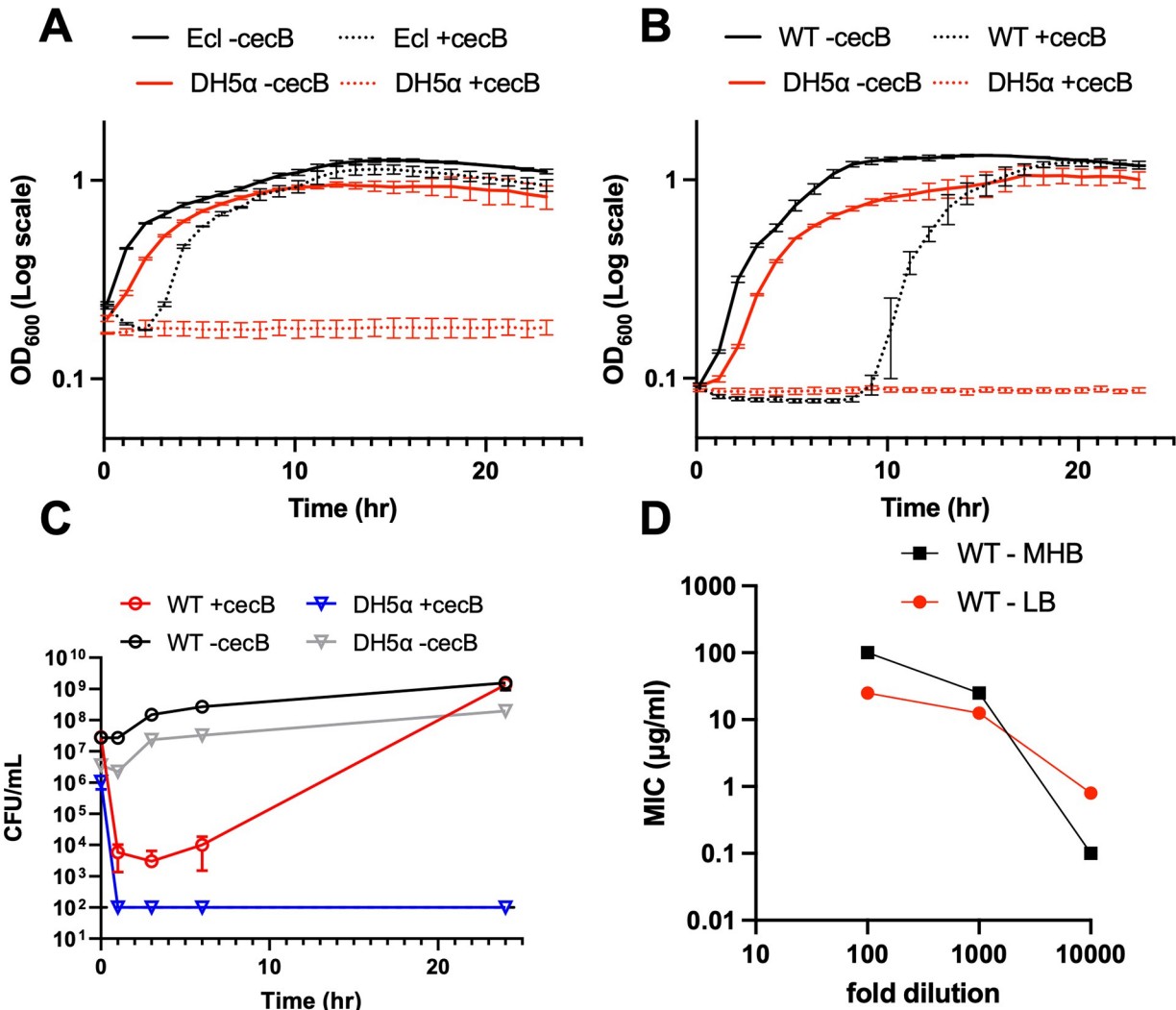

**Fig 1. *E. cloacae* displays inoculum-dependent dynamic survival in the presence of cecB.** Overnight cultures were diluted **(A)** 10-fold or **(B)** 100-fold into fresh LB containing 20 µg/mL cecB, and cells were grown at 37˚C for 24 hours. $OD_{600}$ measurements were taken every 5 minutes. Error bars represent standard deviation (n = 3). **(C)** Cultures were prepared as in (B), but samples were taken at indicated timepoints to quantify colony forming units (CFU) per mL. Error bars represent standard deviation (n = 3). **(D)** Inoculum effect in different growth media. MIC assays were conducted at the indicated dilution of an overnight culture of *E. cloacae*.

opportunistic pathogens, we assessed the killing dynamics of *Enterobacter cloacae* exposed to cecB. We used the well-characterized type strain ATCC 13047 (originally isolated from spinal fluid) as the model of choice. To this end, we treated a high density of cells (10-fold diluted subculture from overnight, about $10^8$ CFU/mL) with cecB in a growth curve assay at 20 µg/mL (2x MIC) (**Fig 1A**). Despite the cecB concentration above the MIC, *E. cloacae* eventually grew after a lag phase of 2 hours. This phenotype, however, was highly dependent on initial cell density. At a more dilute, 100-fold subculture, *E. cloacae* exhibited a longer, but variable apparent lag phase lasting approximately 10 +/- 5 hours (>6 biological replicates; one example shown in **Fig 1B**), followed by exponential growth (**Fig 1B**). In contrast, the laboratory *E. coli* strain DH5α (a strain commonly used in AMP studies [30]) failed to grow in the presence of cecB, despite its MIC being in essential agreement with the *E. cloacae* MIC. We also tested cecB activity against two multidrug resistant *Enterobacter cloacae* complex (ECC) strains encoding

the carbapenemases NDM-1 or NDM-5. These strains are also resistant to other classes of antibiotics such as aminoglycosides, cephalosporins, fluoroquinolones and rifampicin. We observed a similar trend where both NDM-1 and NDM-5 ECC strains exhibited an extended lag phase (~6 hours) followed by exponential growth (**S1A and S1B Fig**). This was again dependent on the initial inoculum as 100-fold subcultures of both NDM-1 and NDM-5 ECC strains were susceptible to 20 μg/mL cecB (**S1C and S1D Fig**). Interestingly, in contrast to ATCC13047, both MDR strains were eradicated at the higher dilution, emphasizing the potential for cecB-like peptides to suppress MDR infections. We next determined viability in cecB-treated cultures. Viable cell counts in the 1:10 dilution dropped up to 10-fold upon exposure to 2x MIC cecB (**S1E Fig**), before recovering, consistent with our $OD_{600}$ measurements. In contrast, upon 1:100 back-dilution, viable cell count dropped ~1000 fold after exposure to cecB before recovering to control levels over a 24-hour period (**Fig 1C**). This was also reflected in MIC measurements where, consistent with previous studies [54–56], we observed a marked inoculum effect, both in LB medium and the standard clinical microbiology medium MHB (**Fig 1D**). Given this effect, "fold-MIC" always refers to the 1000-fold dilution condition in LB in the following, for simplicity's sake. Taken together, our data show that *E. cloacae* exhibits inoculum-dependent killing by an AMP, and a complex dynamic of killing vs. regrowth.

## Delayed growth is due to dynamic survival within a population

To dissect the intriguing observation of delayed growth at high dilution further, we first asked whether regrowth reflected stable genetic suppressor mutations. To this end, we sampled cultures of *E. cloacae* that had been exposed to 2x MIC at the endpoint (overnight post-exposure regrowth) and repeated the growth curve experiment at a 100-fold back dilution with another dose of 2x MIC cecropin (**S2A Fig**). Time to growth was remarkably variable, even across technical replicates of the same biological sample, ranging from 8 to 18 hours. More importantly, however, none of the re-exposed cultures grew immediately upon renewed exposure to cecB, excluding stable resistance as an explanation for the regrowth. Instead, the delayed apparent cecB resistance is transient and stochastic. These phenotypes are reminiscent of "heteroresistance", which is high and transient resistance of a small subpopulation against clinically used AMPs such as the polymyxins [51,57], and also reminiscent of the phenomenologically similar "dynamic persistence" in response to antibiotics, where apparent population stability is mediated by an equilibrium of growth and death of bacterial subpopulations [58].

To better understand the dynamics of *E. cloacae* growth in cecB within a population, we turned to single-cell analysis conducted via phase-contrast microscopy. Cells were grown to mid-exponential phase, then placed on 0.8% agarose LB pads containing cecB and incubated at 37°C while images were acquired. At 2x MIC, 39% of cells immediately began dividing, while 61% immediately lysed (**Fig 2A and 2B**). For comparison, 97% of cells grew when no drug was present. At 1x MIC, 89% of cells immediately divided, and at 20x MIC, 0% of cells divided. Thus, *E. cloacae* populations contain a phenotypically resistant subpopulation of cells that can survive lower cecB concentrations but are killed at higher concentrations. Note that this is different from classical heteroresistance to the therapeutic AMP colistin, where a subpopulation is transiently resistant to even high concentrations of the antimicrobial [57,59]. To evaluate the concentration-dependence more quantitatively, we next conducted lysis experiments by growing *E. cloacae* to mid-exponential phase followed by addition of increasing concentrations of cecB. We observed increasingly delayed growth with increasing cecB concentrations, with near complete lack of recovery at the highest concentrations (**S2B Fig**). Taken together, these data strongly suggest that regrowth observed in our killing experiments is caused by initial cecB dynamic survival/dynamic persistence (cycles of growth and death)

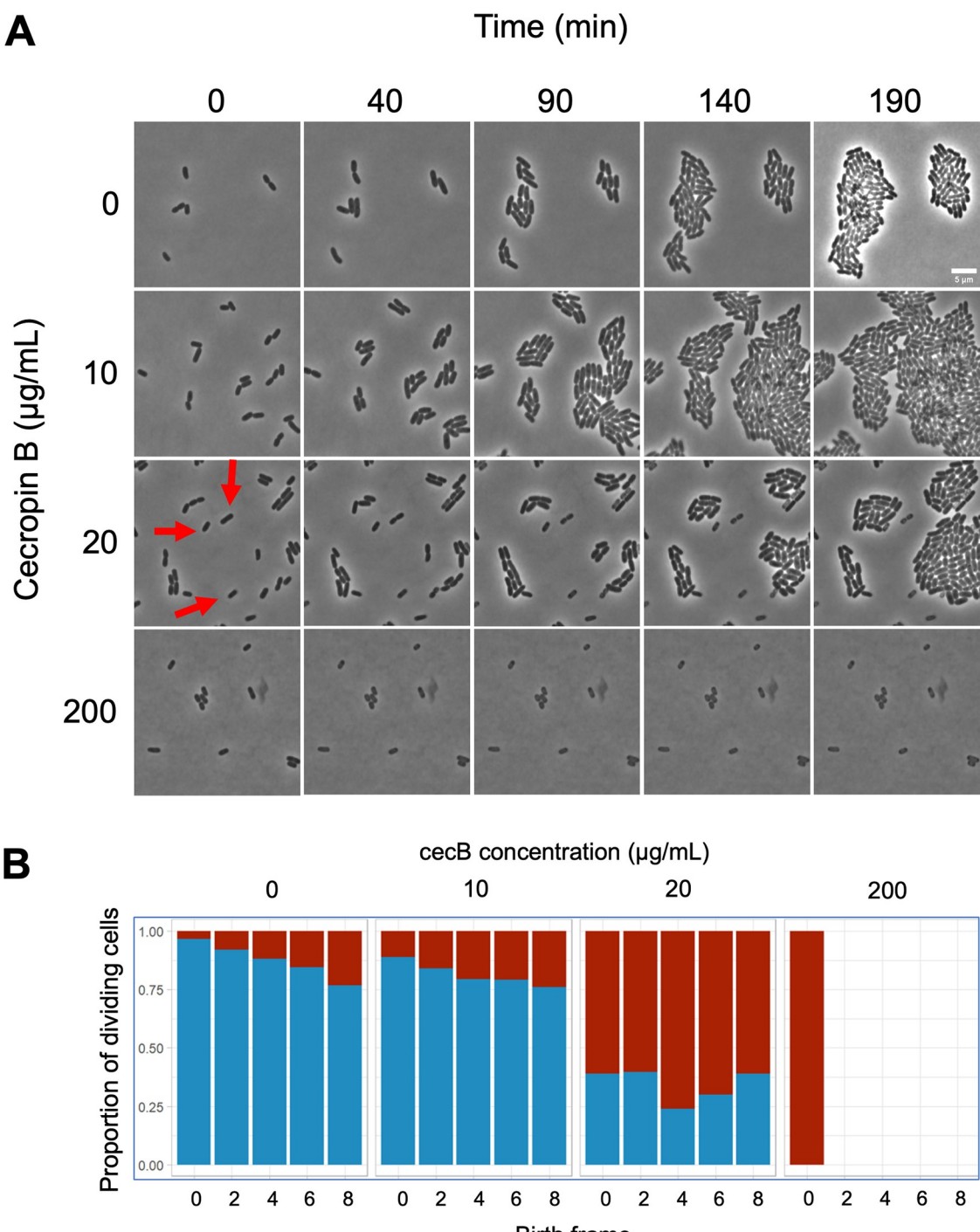

**Fig 2. *E. cloacae* exhibits population heterogeneity in its response to cecropin B. (A)** Cells were grown to mid-exponential phase in liquid medium and then placed on 0.8% agarose pads containing the indicated amounts of cecropin and incubated at 37°C. Representative, cropped frames from a time series are shown. In the 20 µg/mL condition, red arrows indicate examples of lysing cells. **(B)** Percentage of cells that divided during the time lapse, grouped by drug concentration (0, 10, 20, 200 µg/mL). Cells are binned by birth frame, i.e., the frame of the time lapse during which the cell was first identified by SuperSegger [85]. Blue represents cells that underwent a successful division event throughout the course of the timelapse, red indicates cells which did not divide. Missing bars indicate that the initial population of cells was unable to divide.

but not primarily by hypertolerant persister cells that are refractory to antimicrobial killing independent of compound concentration [60–62]).

We then asked whether regrowth might be caused by a transient physiological adaptation to the AMP, e.g. via induced OM modifications. If so, we would predict that the emerging growing population should be more refractory to subsequent exposure. To test this, we grew *E. cloacae* to mid-exponential phase and exposed the population to 20 μg/mL cecB (2x MIC) at $OD_{600}$ = 0.5. As in previous experiments, the culture density decreased at first (indicating lysis), followed by regrowth. We then exposed the culture to the same concentration of cecB again after reaching $OD_{600}$ = 0.5. Both lysis (**S2C Fig**) and survival after 1 hour (**S2D Fig**) were similar at both exposures, suggesting that cells surviving the first cecB exposure were not primed to survive second exposure. Rather, we propose that the cecropin is either degraded by proteases, or sequestered by dead cells at this point in the experiment (see discussion for details).

## PhoPQ partially contributes to cecB resistance

To investigate the molecular mechanisms underlying dynamic survival in *E. cloacae*, we took a targeted genetic approach. The PhoPQ two-component system is well-characterized as a regulator of AMP resistance mechanisms in Enterobacterales and is essential for colistin heteroresistance [57]. In short, the sensor kinase PhoQ autophosphorylates in response to AMPs and other stimuli ($Mg^{2+}$ deficiency, osmotic stress, pH) [63], resulting in phosphorylation of the response regulator PhoP. Phosphorylated PhoP then upregulates expression of many genes, including those encoding enzymes associated with outer membrane modifications that promote AMP resistance through membrane charge alterations (e.g., the *arn* operon, which modifies LPS with positively-charged L-amino arabinose in *E. cloacae*) [57]. To test the role of PhoPQ in cecB dynamic survival, we constructed a Δ*phoPQ* mutant and tested its resistance to cecB. The Δ*phoPQ* cecB MIC was in essential agreement (2-fold lower) with WT. We then tested the Δ*phoPQ* mutant in a growth curve assay at 2x WT MIC. At a 10-fold back dilution, the Δ*phoPQ* mutant population exhibited delayed growth (preceded by partial lysis) compared to WT (**Fig 3A**), taking 3 hours to enter exponential growth. At a 100-fold dilution, this difference was exacerbated to 8 hours after WT entered exponential growth (**Fig 3B**). Complementation of *phoPQ* via plasmid-based overexpression restored resistance to WT levels (**Fig 3A and 3B**). Viable cell counts of Δ*phoPQ* (when diluted 100-fold into 2x WT MIC cecB) indicated more severe killing than WT initially, but the ability to recover by 24 hr was unaffected (**S1E Fig**), consistent with $OD_{600}$ measurements (**Fig 3B**). Interestingly, we observed a marked condition-dependence of survival dynamics in the mutant. While the Δ*phoPQ* mutant grew at 2x WT MIC after just a few hours delay in a growth curve assay in liquid culture, exposure to 1x WT MIC resulted in complete lack of growth on an agarose pad, with some cells becoming slightly phase light (indicative of cell envelope integrity failures), while others stayed apparently intact, but did not grow (**Fig 3C**). In either liquid culture or agarose pads, however, the Δ*phoPQ* mutant was more cecB sensitive than the WT. These results are consistent with the well-established view that PhoPQ acts as an important AMP resistance mechanism.

## The Rcs phosphorelay partially contributes to AMP resistance

Since we still observed dynamic survival (albeit at a reduced level) in the Δ*phoPQ* mutant, we explored additional putative AMP resistance factors as the underlying cause of this phenotype. The Rcs (regulator of capsule synthesis) system is a cell envelope stress response system present across Enterobacterales. The system comprises the RcsFCDB phosphorelay, which responds to cell envelope stress and effects its response through the transcriptional regulator RcsB and, as one of its main functions, upregulates capsule synthesis [64]. AMPs induce the Rcs system

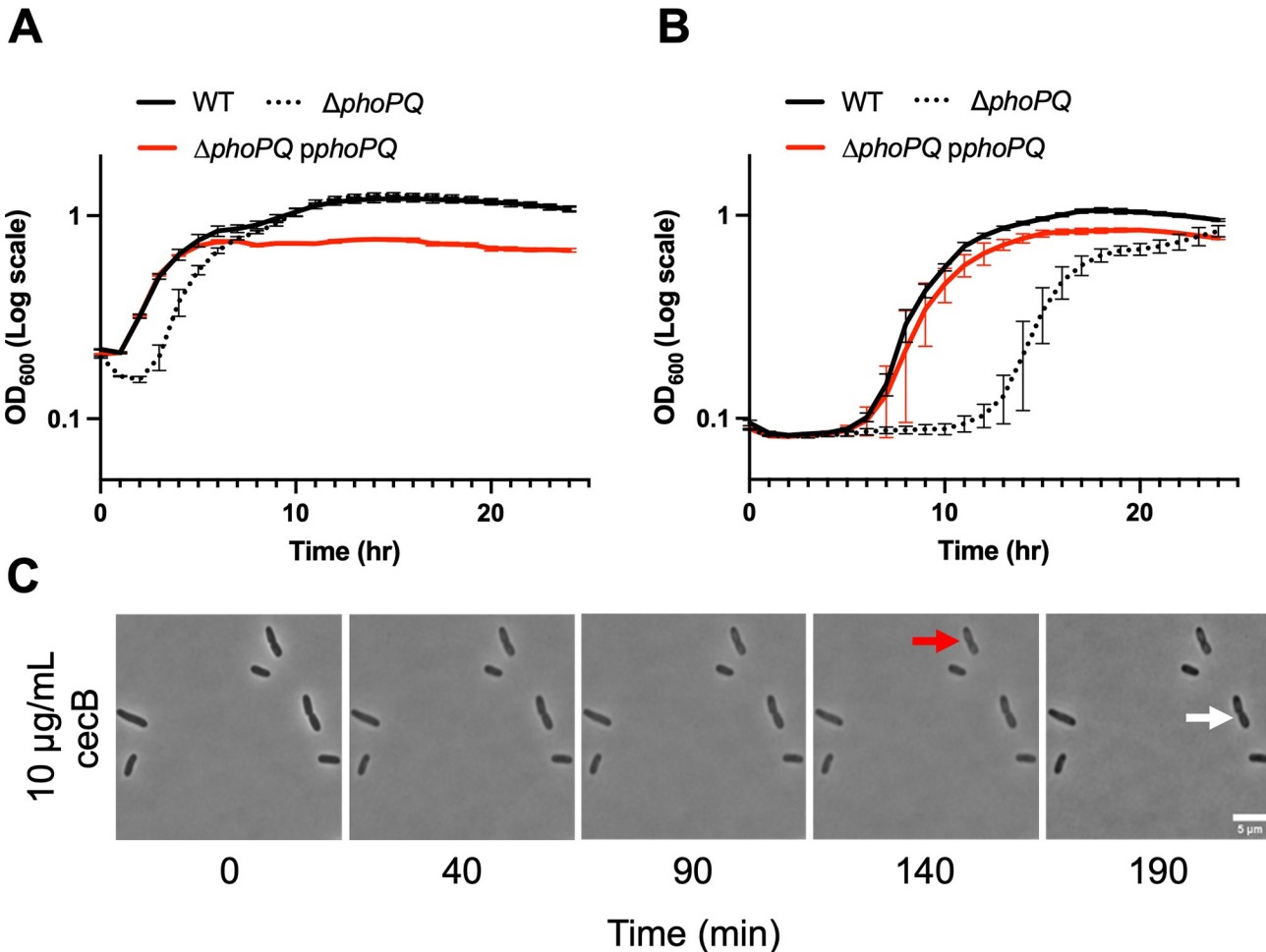

**Fig 3. PhoPQ partially contributes to cecB resistance.** Overnight cultures were diluted **(A)** 10-fold or **(B)** 100-fold into fresh LB containing 20 μg/mL cecB, and cells were grown at 37°C for 24 hours. For the complement strain, expression of *phoPQ* from pMMB was induced with 200 μM IPTG. $OD_{600}$ measurements were taken every 5 minutes. Error bars represent standard deviation (n = 3). **(C)** Mid-exponential phase cells were placed on 0.8% agarose pads containing the indicated amounts of cecropin and incubated at 37°C. Representative, cropped time frames are shown. Red arrow indicates phase-light cell, white arrow indicates a cell that stays intact.

[65], and the colanic acid capsule that this system positively controls has been implicated in AMP resistance in other bacteria [45], though the extent to which Rcs contributes to AMP resistance is largely unknown. We therefore constructed a Δ*rcsB* mutant and conducted time-dependent killing experiments with cecB. At 2x WT MIC, the Δ*rcsB* mutant exhibited reduced survival compared to WT, undergoing initial lysis, followed by growth after 5 hours +/- 2 (10-fold dilution), or 20 hours +/- 4 (100-fold dilution), respectively (**Fig 4A and 4B**). Once again, viable cell counts were consistent with this observation (**S1F Fig**). In timelapse microscopy, Δ*rcsB* was completely unable to grow at 2x WT MIC, a concentration at which 39% of WT cells grow. At 1x WT MIC, 94% of Δ*rcsB* cells grew (**Fig 4C**). Taken together, our data suggest that the Rcs phosphorelay contributes substantially to cecB resistance in *E. cloacae*.

## OmpT protease contributes to cecB resistance via proteolytic cleavage

We then turned our attention to proteases, which are well-characterized as a mechanism of resistance against AMPs [66,67]. We first analyzed the *E. cloacae* genome for homologs of

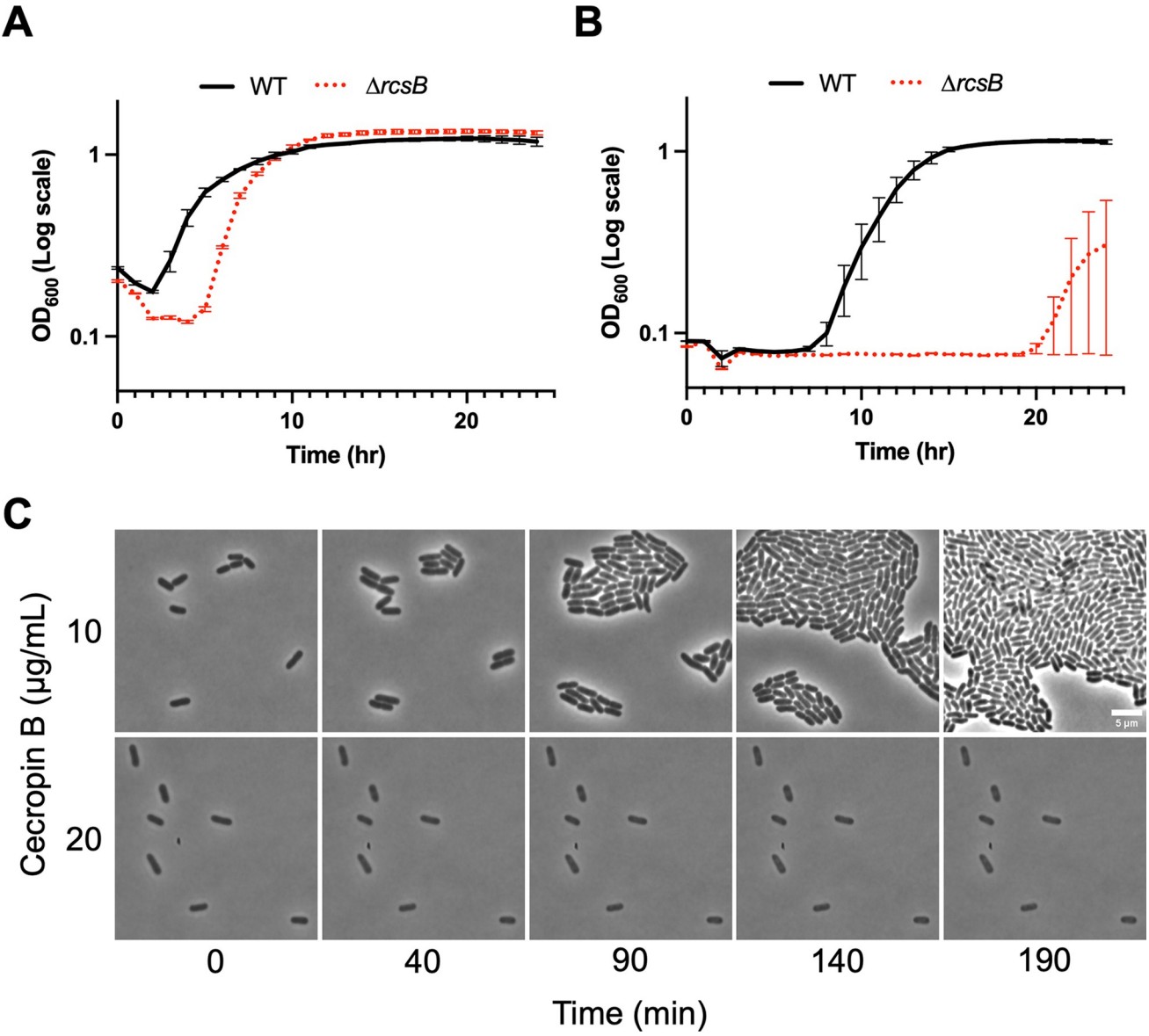

**Fig 4. The Rcs phosphorelay partially contributes to resistance.** Overnight cultures were diluted **(A)** 10-fold or **(B)** 100-fold into fresh LB containing 20 µg/mL cecB, and cells were grown at 37°C for 24 hours. $OD_{600}$ measurements were taken every 5 minutes. Bars at each point represent standard deviation (n = 3). **(C)** Mid-log cells were placed on 0.8% agarose pads containing indicated amounts of cecropin and incubated at 37°C. Representative, cropped frames from a timelapse movie are shown.

proteases that have been implicated in AMP degradation. We identified homologs of *Serratia marcescens prtS* (73% AA identity), *E. coli ompT* (55% AA identity) (**S3 Fig**), and two homologs of the *E. coli* Sap system (38% and 36% AA identity). The Sap system, which imports AMPs for cytoplasmic degradation, has been implicated in AMP resistance for various Gram-negative pathogens [41,68]. PrtS is a zinc metalloprotease that cleaves cecropin A [69], while OmpT is a well-characterized *E. coli* outer membrane protease, which, in addition to housekeeping functions [70], degrades the human AMP LL-37 (cathelicidin) and thereby promotes intestinal colonization [71].

To test the impact of these putative proteases on *E. cloacae* resistance to cecB, deletion mutants were generated. Of the three single mutants, only deletion of *ompT* caused increased susceptibility to cecB in liquid culture at 10- and 100-fold dilutions at 2x WT MIC (**Figs 5A and S4**). The *ΔompT* mutant underwent lysis before regrowth at 6 hours (+/- 3) at a 10-fold dilution, or 13 hours (+/- 4) at a 100-fold dilution (**Fig 5A and 5B**). Overexpression of *ompT in trans* not only complemented the phenotype but resulted in increased resistance compared to WT, suggesting an important role for OmpT in resistance to cecB (**Fig 5A and 5B**). In single cell analysis at 1x WT MIC, 40% of *ΔompT* mutant cells grew while the remaining cells in the population lysed. Notably, descendants of cells that initially displayed resistance frequently lysed at later timepoints (**Fig 5C**), consistent with the notion that cecropin dynamic survival is a transient state and not a defined, homogeneous phenotype across a population.

We next aimed to disrupt the proteolytic function of OmpT to ask if the predicted active site was required for OmpT-mediated cecB resistance. Four amino acid residues, a His-Asp and Asp-Asp couple, have been implicated in the OmpT active site in *E. coli* [72]. These residues are conserved in *E. cloacae* OmpT (D104, D106, D226, and H228). Due to the close proximity of residues, two amino acid substitution mutant strains were created: OmpT$^{D104A/D106A}$ and OmpT$^{D226A/H228A}$. Disruption of either AA couple abrogated the ability of plasmid-borne OmpT to rescue cecB sensitivity of the *ΔompT* mutant (**Fig 5D**), suggesting OmpT proteolytic activity is important for *E. cloacae* survival in the presence of cecB.

We were intrigued by the involvement of OmpT, since *E. coli* DH5α encodes an OmpT homolog (50% AA identity) but was not resistant against cecB. We thus tested whether *E. cloacae ompT* expression conferred cecB resistance to DH5α. OmpT$_{Ecl}$ overexpression was sufficient to promote resistance to cecropin in DH5α, which otherwise is completely susceptible to 2x WT MIC levels of cecropin (**S5A Fig**). We were also curious whether lack of *E. coli* OmpT proteolytic activity was due to low native expression levels. Overexpression of OmpT$_{DH5α}$ in an *E. coli* DH5α background provided some resistance against cecB where the cells resumed exponential growth after a long lag phase (**S5B Fig**). However this was not sufficient to fully restore the resistant phenotype observed with overexpression of OmpT$_{Ecl}$ in *E. coli* DH5α background. This suggests that *E. cloacae's* native OmpT cleaves cecB while *E. coli* OmpT shows limited activity against cecB, even when expressed at higher than native levels.

## PhoPQ, Rcs, and OmpT are collectively required for cecB resistance

Individual deletions of *phoPQ*, *rcsB*, or *ompT* resulted in a significant increase in susceptibility to cecB; however, all mutants still exhibited (albeit, reduced) dynamic survival around WT MIC concentrations. Notably, repassaging the resistant mutant populations into fresh cecB resulted in sensitivity comparable to the initial exposure in all mutants, confirming that the populations do not harbor stable genetic mutations and suggesting that cecropin may have been depleted from the media (**S6 Fig**). To test the additive effect of these deletions, double and triple knockout strains were constructed, starting with a *ΔphoPQ* background. Upon exposure to 2x WT MIC at a 10-fold or 100-fold dilution, the *ΔphoPQ ΔompT* double mutant had a more pronounced growth delay than either single mutant (**S7A Fig**), suggesting these mechanisms independently promote dynamic survival in the presence of cecropin. Similarly, a *ΔphoPQ ΔrcsB* double mutant had a more pronounced growth delay than either *ΔphoPQ* or *ΔrcsB* mutants (**S7A Fig**). It should be noted that some replicates of *ΔphoPQ ΔrcsB* failed to grow at all in the presence of cecB, but this was not frequently observed (**S7B Fig**).

In contrast, deletion of PhoPQ, OmpT, and RcsB (*ΔphoPQ ΔompT ΔrcsB*, "Δ3") resulted in near complete eradication by cecropin at both 10-fold and 100-fold back dilution (**Fig 6A and 6B**), with no full regrowth at least up to 24 hours. In almost all experiments, the triple mutant

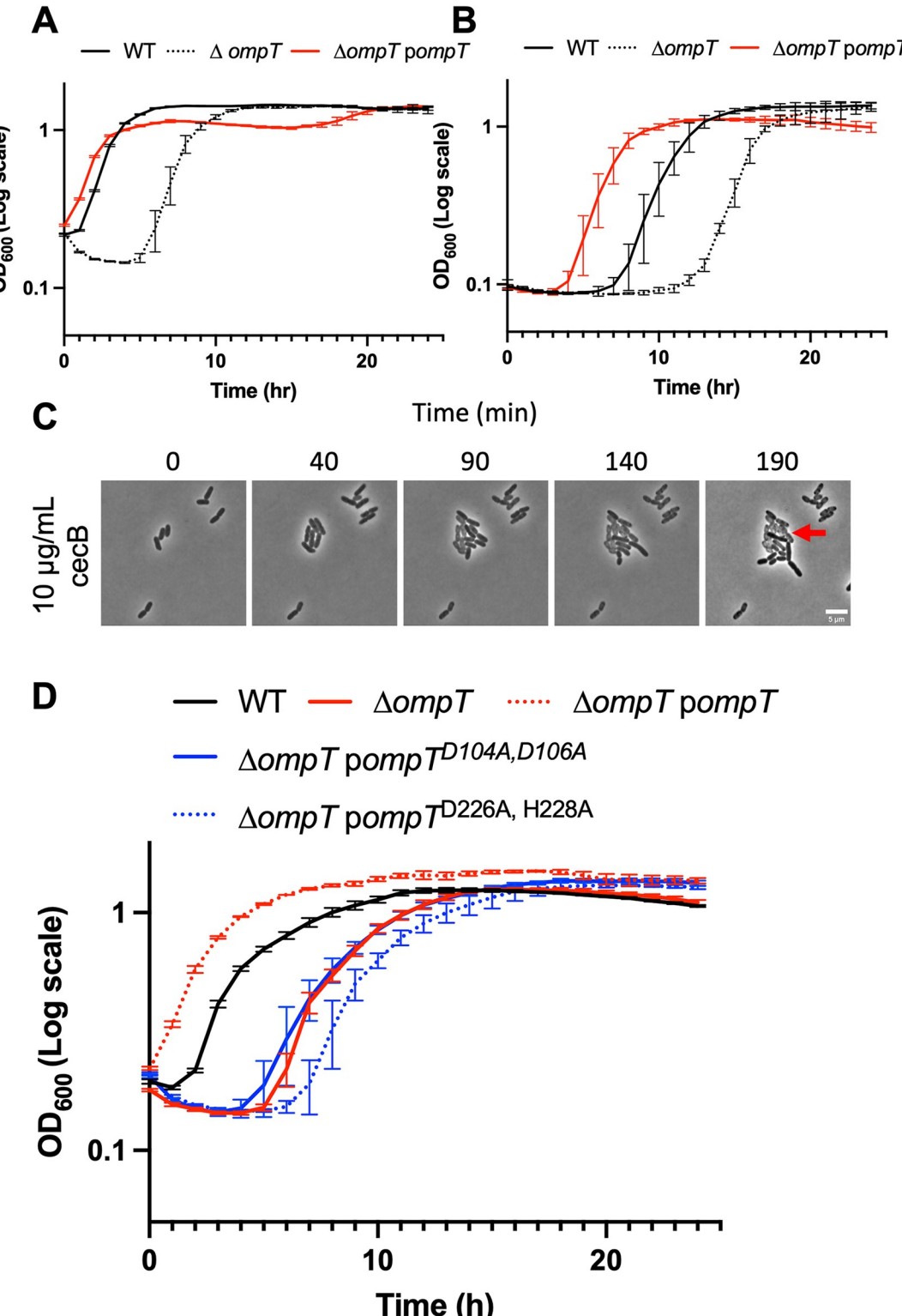

**Fig 5. OmpT contributes to cecB resistance via proteolytic cleavage of cecB.** Overnight cultures were diluted (**A**) 10-fold or (**B**) 100-fold into fresh LB containing 20 μg/mL cecB, and cells were grown at 37°C for 24 hours. Where applicable, expression of *ompT* from pBAD was induced with 0.1% arabinose. $OD_{600}$ measurements were taken every 5 minutes. Error bars represent standard deviation (n = 3). (**C**) Mid-exponential cells were placed on 0.8% agarose pads containing the indicated amounts of cecropin and incubated at 37°C. Representative, cropped frames from a timelapse movie are shown. Red arrow indicates a cell

which initially divided, but later succumbed to cecropin and lysed. **(D)** Effects of OmpT active site mutations were tested using conditions in **(A)**.

was completed killed by the AMP (**Fig 6C**), though we did note a small surviving fraction in one experiment (perhaps indicating that some rare AMP persisters exist [48]). In addition, single cell analysis revealed that Δ3 was completely lysed by cecropin at 1x WT MIC (**Fig 6C and 6D**), though at this lower concentration, we observed initial attempts at a few divisions on the agarose pad before lysis. Despite this dramatic phenotype, however, we still observed an inoculum effect in MIC measurements for Δ3 (**S8 Fig**)–while the absolute values for MICs were lower than for WT, the dynamic relationship between inoculum size and MIC was preserved. In aggregate, our data show that all three AMP resistance mechanisms collectively contribute to cecB resistance in *E. cloacae* and inhibition of all 3 AMP resistance mechanisms could lead to almost total elimination of cecB resistance.

## Discussion

As pathogens continue to develop resistance against antibiotics at an alarming rate, AMPs have come into focus as potentially safe and effective therapeutics with which to treat MDR infections [73–76]. With many AMPs (particularly including cecropins [24,25]) undergoing evaluation as clinically useful antimicrobials, it is imperative to determine how bacteria sense and respond to AMP treatment, potentially enabling the development of pre-emptive measures against inevitable resistance development. While classical antibiotic resistance can often be pinned to a single genetic factor (e.g., target modification [77], drug inactivation [78] and efflux [79]), we demonstrate that AMP resistance can be far more nuanced. In the case of cecropin, we show that multiple resistance strategies are deployed by *Enterobacter cloacae*– outer membrane modifications, protease production, and membrane stress response collectively maximize *E. cloacae*'s survival in the presence of cecB (**Fig 7A**). Thus, *E. cloacae* is not entirely dependent on any one of these factors to survive cecropin exposure, and, in liquid culture, presence of just one was sufficient to promote resistance. Since cecropin mimics the mechanism of action (LPS disruption) of many human AMPs, it is likely that the resistance mechanisms studied here have broad implications for pathogenesis.

We also observed "dynamic survival" (reminiscent of heteroresistance) to cecropin. Resistance was unstable, with populations rapidly reverting to initial cecB sensitivity upon additional doses of drug. One potential explanation for lack of stable cecropin resistance is the formation of unknown tandem gene amplifications, which are intrinsically unstable and thus transient [80]. More likely, the phenotype we observed could be due to stochastic variance of gene expression in cells, even across a genetically identical population. For example, if 50% of cells in the population are expressing an initial "optimal" level of OmpT or PhoPQ-regulated genes, they will survive and divide, while the remaining cells die. Without antimicrobial pressure, heterogeneity of gene expression may return to the population. This is supported by the fact we observed individual cells in populations of *E. cloacae* displaying seemingly stochastic and divergent outcomes. Regardless of the mechanism, this striking population heterogeneity speaks to the importance of single-cell analysis when investigating bacterial susceptibility to AMPs.

It is also intriguing that even a 10-fold difference in initial inoculum resulted in drastically different outcomes at the population level. An inoculum effect has been demonstrated previously with the conclusion that, as a universal property of AMPs, a certain AMP/cell ratio must be reached for killing to occur [56]. This is particularly important for AMPs compared to classical antibiotics, given that AMPs have a very low range of intermediate efficacy [8]. In other

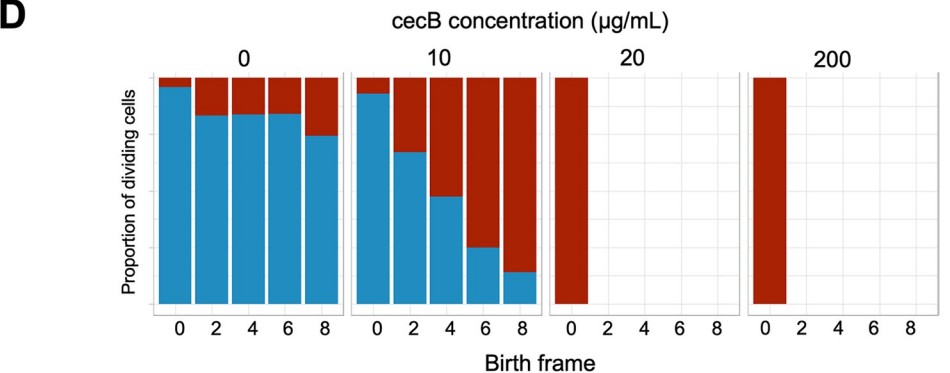

**Fig 6. PhoPQ, Rcs, and OmpT are collectively required for cecB resistance.** Overnight cultures were diluted **(A)** 10-fold into fresh LB containing 20 μg/mL cecB, and cells were grown at 37°C for 24 hours. OD$_{600}$ measurements were taken every 5 minutes. Bars at each point represent standard deviation (n = 3). **(B)** Cultures were diluted 100fold into fresh medium with or without cecropin; samples were taken at indicated timepoints to quantify colony forming units (CFU) per mL. Error bars represent standard deviation (n = 3). **(C)** Mid-exponential phase cells were placed on 0.8%

agarose pads containing indicated amounts of cecropin and incubated at 37˚C. Representative, cropped time frames are shown. **(D)** Percentage of cells that divided during the time lapse, grouped by drug concentration (0, 10, 20, 200 μg/mL). Cells are binned by birth frame, i.e., the frame of the time lapse during which the cell was first identified by SuperSegger [85]. Blue represents cells that underwent a successful division event throughout the course of the timelapse, red indicates cells which did not divide. Missing bars indicate that the initial population of cells was unable to divide.

words, AMPs will have little effect on a bacterial cell up until reaching an AMP/cell ratio where complete killing occurs. Given this narrow range, it is plausible that an order of magnitude more cells in a population can afford a tremendous growth advantage to *E. cloacae* treated with cecropin. The dynamics of this ratio become more complex when considering an actively growing single cell. For instance, as AMPs are binding to a cell and nearing the critical killing threshold, what if the cell completes its division? Suddenly, the AMP/cell ratio has been halved (at least in a simple model; heterogeneous distribution of surface-bound AMPs among daughter cells is also conceivable)–maximal AMP killing may thus be a function of both AMP concentration and bacterial division rate. In this way, the binding kinetics of a particular AMP may be vital in its ability to kill a population of bacteria. Further, dead cells within a population can play a protective role to growing and dividing cells; the membranes of lysed cells can absorb AMPs which could otherwise act on growing bacteria [81,82]. We propose that each of the genetic factors we describe in this study help *E. cloacae* to raise the AMP/cell ratio necessary to lyse a cell. In WT, for instance, although some cells will initially die, the vast majority of cells in the population will not be bound by enough cecB to lyse, due to degradation of some surface-bound cecB (via OmpT), reduction of target binding (through PhoPQ-mediated OM modifications), and/or delay in binding due to (putatively) capsule production (induced by Rcs). However, even at higher AMP concentrations, all it takes is for a few cells to win the initial race of division vs AMP binding for the population to eventually recover, if the AMP concentration sinks to growth-permissive levels in the meantime. Sequestration by dead cells or slow degradation by another protease therefore likely contributes to allowing regrowth to full density after 24 hours in our experimental stup. We propose a model wherein the eventual survival of an *E. cloacae* population is dependent on the ability of the population to lower the cecB concentration to beneath its effective killing threshold (**Fig 7B**). In the case of our Δ3 mutant, the population lyses completely before cecropin levels have been sufficiently lowered. This model would also explain why the resulting population is sensitive to cecropin if exposed again, because no individual cell in the population exhibits outright resistance, and is consistent with our observation that the Δ3 mutant still exhibits an inoculum effect, albeit at lower AMP concentration.

AMPs are incredibly diverse in structure, mode of action, and efficacy against human pathogens [13]. Specific AMP resistance mechanisms may therefore not be equivalent across bacteria. For instance, DH5α encodes its own homologue of OmpT, which appeared to be fairly ineffective in combating cecB compared to *E. cloacae's* OmpT, even when overexpressed. Indeed, the surprising diversity of OmpT-mediated AMP degradation, even within the same species, has been noted before in different strains of *E. coli* [71].

Altogether, this work demonstrates how multiple genetic factors can independently contribute to AMP resistance in an opportunistic pathogen. This supports the notion that transient, stochastic resistance, or "dynamic survival" may have an impact on infection outcomes. Further study will be necessary to parse out the fine mechanisms which govern the ever-evolving battle between host immunity and pathogenic invaders.

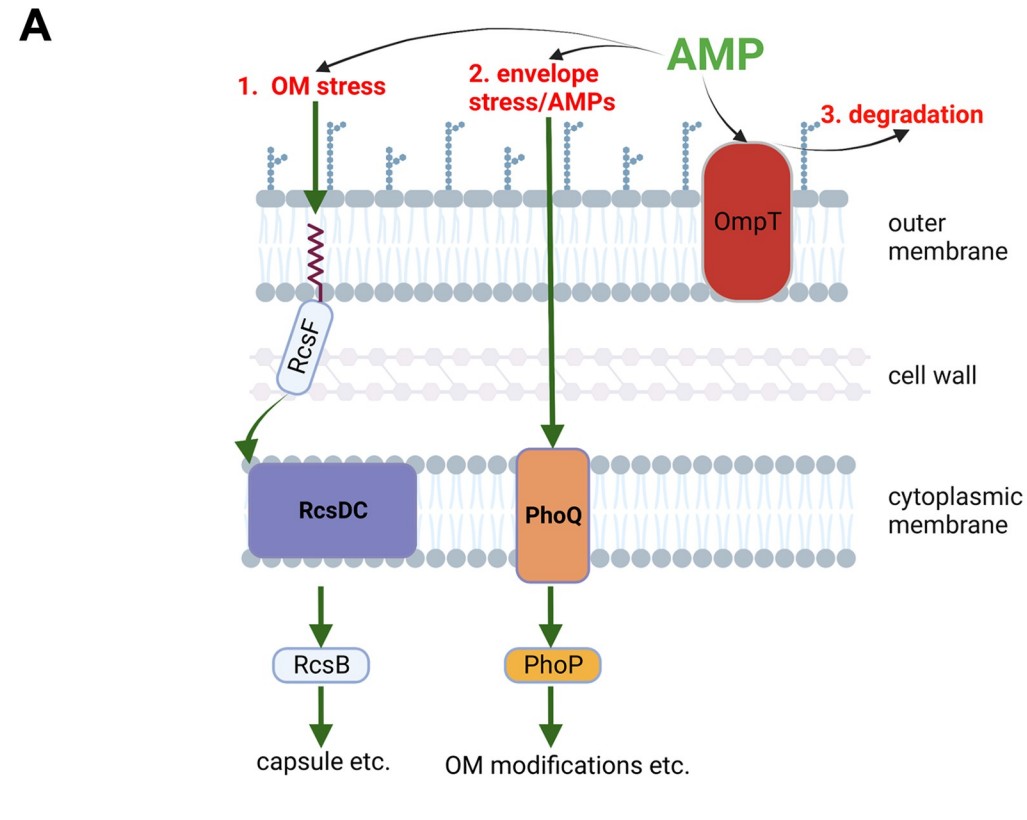

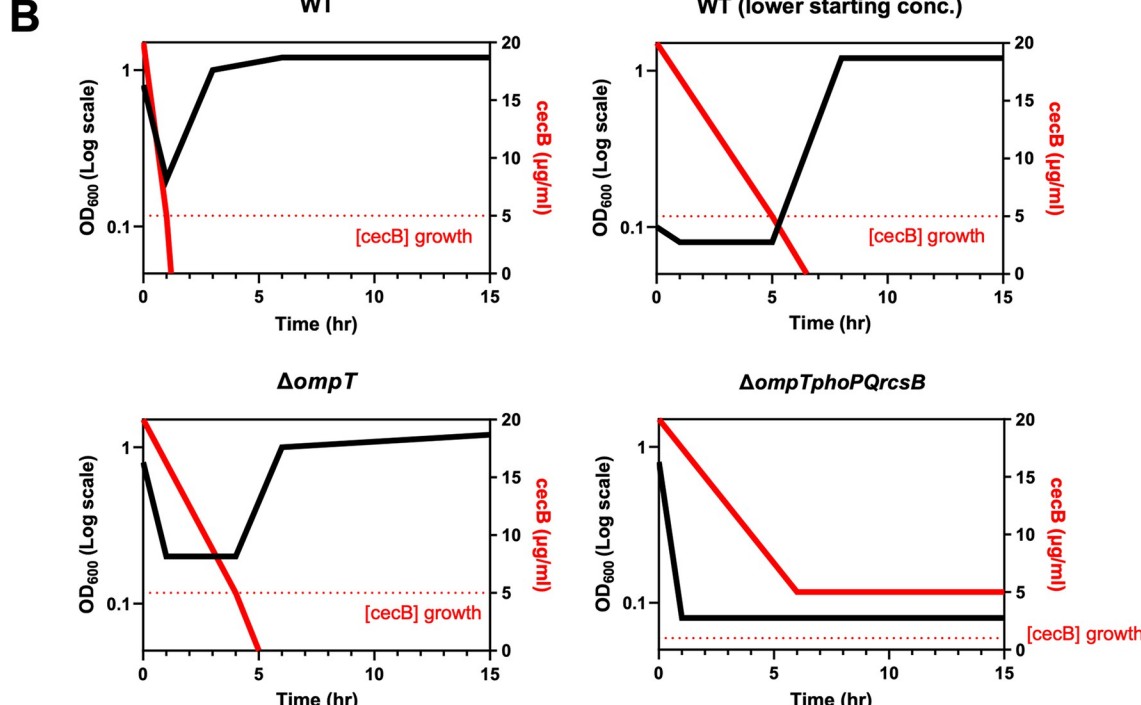

**Fig 7. Model of *Enterobacter cloacae* cecropin B dynamic survival. (A)** Mechanisms of AMP resistance discussed in this study. 1) OM stress induces the Rcs phosphorelay, which phosphorylates the response regulator RcsB. Among other systems, RcsB upregulates capsule production. 2) AMPs and envelope stress induces the PhoPQ signaling system, which upregulates transcription of enzymes which modify the outer membrane. 3) OmpT, bound in the outer membrane, may be able to cleave AMPs before they act on the cell surface. **(B)** Model of *E. cloacae* growth vs concentration of cecB over time. Black line represents hypothetical population density

(indicated by $OD_{600}$), solid red line represents the hypothetical concentration of cecropin in the sample, and the dotted red line represents the hypothetical concentration of cecropin necessary for lysis. In this model, the population must bring the amount of cecropin below the effective killing concentration before all cells in the population are lysed. Factors like initial starting concentration or protease production may modify the rate at which bacteria can reduce cecropin concentration, while other factors like PhoPQ and RcsB may raise the concentration of cecropin necessary for lysis.

## Methods

### Bacterial strains and growth

All strains, plasmids, and primers used in this study are listed in S1 and S2 Tables. We used the *Enterobacter cloacae* type strain ATCC13047 and its derivatives for all experiments. All strains were initially grown from freezer stocks on solid Luria-Bertani (LB) agar at 37°C. Isolated colonies were used to inoculate LB at 37°C. Where applicable, kanamycin was used at 50 μg/mL, chloramphenicol at 100 μg/mL, and cecropin B (MedChemExpress, Monmouth Junction, NJ) at 20 μg/mL unless otherwise noted. For complementation experiments, 0.2% arabinose or 200 μM Isopropyl-ß-D-thiogalactopyranoside (IPTG) was used to induce plasmid-borne genes. *E. cloacae* NDM-1 and NDM-5 were obtained from the culture collection of Weill Cornell Medical (S3 Table).

### Mutant construction

The *ompT*, *rcsB*, *sapA*, and *prtS* mutants were constructed using the suicide vector pTOX5 as described in [83]. ~500 bp upstream and downstream flanking homology regions were amplified from ATCC13047 using primers ANM28/29 and ANM30/31 for *ompT*, ANM78/79 and ANM80/81 for *rcsB*, ANM16/17 and ANM18/19 for *sapA*, and ANM42/43 and ANM44/45 for *prtS*. For a given gene, flanking homology regions were then cloned into pTOX5 digested with EcoRV using isothermal assembly [84]. Successful constructs were then transformed into *E. coli* donor strain MFDλ*pir* and subsequently conjugated into ATCC13047. Recombinants were selected on LB plates containing 1% glucose and 100 μg/mL chloramphenicol. Upon single colony purification, colonies were directly streaked out on an M9 minimal medium plate containing 0.2% casamino acids and 1% rhamnose, followed by incubation at 37°C for 24 hours. Mutants were then tested with flanking primers for each gene (*ompT*, ANM26/27; *rcsB*, ANM86/87; *sapA*, ANM14/15; and *prtS*, ANM32/33).

Amino acids substitutions in *ompT* were generated using the Q5 Site-Directed Mutagenesis Kit (New England Biolabs, Ipswich, MA). Briefly, primers containing the desired mutations were designed (D104A-D106A, ANM146/147; D226A-H228A, ANM148/149) to anneal to *ompT* cloned onto the pBAD33 vector and amplify the entire construct. Kinase, Ligase, and DpnI (KLD) reaction mix was then used to phosphorylate the ends of the PCR products and ligate them. The resulting mutant *ompT* constructs were sequenced using pBAD flanking primers (TC34/35) and electroporated into Δ*ompT*.

### Growth curve experiments

Growth curve experiments were conducted using 100-well honeycomb plates in a Bioscreen C growth curve analyzer (Growth Curves USA, Piscataway NJ). Overnight cultures grown at 37°C (shaking 200 rpm) were diluted 10- or 100-fold into fresh LB medium containing cecropin B (20 μg/mL, 2x MIC) and transferred to honeycomb plates (200 μL volume/culture). Cultures were then grown at 37°C $OD_{600}$ was measured automatically by the plate reader every 10 minutes for 24 hours.

## Cecropin killing experiments

Overnight cultures were grown at 37˚C shaking (200 rpm). For the experiment, cultures were then diluted 10 or 100-fold into fresh LB medium containing cecropin B (20 µg/mL, 2x MIC) in 1.5 mL microfuge tubes. Subcultures were incubated in a 37˚C standing incubator. At indicated timepoints, a 50 µL aliquot was removed from the sample and serially diluted to determine CFU/mL.

## MIC assays

Cultures were grown overnight at 37˚C shaking, then diluted 1000-fold into fresh LB. Subcultures were grown for 1 hour at 37˚C shaking before being diluted 1000-fold again into fresh LB to create a "seed culture". One-hundred microliters of seed culture was subsequently diluted 2-fold into a 96-well plate containing cecropin B concentrations ranging 0.13–64 µg/mL. Reported values are medians of 4 technical replicates.

## Microscopy

Cells were harvested in mid-exponential phase and spotted on a 0.8% agarose pad containing LB and indicated concentrations of cecropin B. Cells were kept at 37˚C while being imaged on a Leica Dmi8 inverted microscope. Images were acquired every 10 minutes. Time lapse images were analyzed using SuperSegger [85] and custom R scripts. Errors were excluded from the analysis by manual curation.

## Supporting information

**S1 Fig. PhoPQ, RcsB, and OmpT each promote *E. cloacae* resistance to cecropin.** Overnight cultures of MDR ECC encoding NDM-1 or NDM-5 were diluted 10-fold **(A and B)** or 100-fold **(C and D)** into fresh LB containing 20 µg/mL cecB, and cells were grown at 37˚C for 24 hours. $OD_{600}$ measurements were taken every 10 minutes. Error bars represent standard deviation (n = 6). Overnight cultures of WT and **(E)** Δ*phoPQ* **(F)** Δ*rcsB* or **(G)** Δ*ompT* were diluted 10-fold into fresh LB containing 20 µg/mL cecB, and cells were grown at 37˚C for 24 hours. Samples were taken at select timepoints to quantify colony forming units (CFU) per mL. Error bars represent standard deviation (n = 3). In (C), individual replicates of Δ*ompT* +cecB are shown due to high variance at the 3-hour timepoint.
(TIF)

**S2 Fig. (A)** WT cells from three independent wells in Fig 1A were sampled after 24 hours of growth in the presence of cecropin, grown overnight in LB, then re-exposed to 20 µg/ml cecB after 100fold dilution. Each of the 3 biological replicates is represented by a different color (black, red, blue), and each color contains 18 technical replicates. **(B)** Concentration-dependent growth and lysis. Overnight cultures were diluted 100fold into fresh medium containing the indicated concentration of cecB, growth ($OD_{600}$) was measured in a plate reader. **(C)** Overnight cultures were diluted 100fold into fresh medium, grown to OD = 0.5, then cec B (2 x MIC) was added. Following regrowth to the same OD, cecB was again added at the same concentration **(D)** Cultures were treated as described in (C), but plated for CFU/mL after 1 hour of exposure to cecB.
(TIF)

**S3 Fig. The structure of *E. cloacae* OmpT is conserved with *E. coli* OmpT.** The crystal structure of *E. coli* OmpT (blue), overlaid with an AlphaFold prediction of the homologous OmpT we identified in *E. cloacae* (red). Pairwise amino acid sequence alignment revealed 50%

identity.
(TIF)

**S4 Fig. Homologues of the Sap system and protease PrtS do not promote cecropin resistance in *E. cloacae*.** Overnight cultures were diluted 10-fold into fresh LB containing 20 μg/mL cecB, and cells were grown at 37˚C for 24 hours. Error bars represent standard deviation (n = 3).
(TIF)

**S5 Fig. *E. cloacae* OmpT confers cecropin resistance to DH5α.** Overnight cultures were diluted 10-fold into fresh LB containing 20 μg/mL cecB, and cells were grown at 37˚C for 24 hours. Error bars represent standard deviation (n = 3).
(TIF)

**S6 Fig. CecB resistance in mutant populations is unstable.** Δ*phoPQ*, Δ*rcsB*, and Δ*ompT* were grown at 37˚C in the presence of 20 μg/mL cecropin. After 24 hours, surviving cells were passaged ON in LB, then diluted 100-fold and re-exposed to 20 μg/mL cecropin. Cells were grown at 37˚C for 24 more hours. Error bars represent standard deviation (n = 3).
(TIF)

**S7 Fig. Deletion of *ompT* or *rcsB* in a Δ*phoPQ* background exacerbates cecropin sensitivity.** Overnight cultures were diluted 10-fold into fresh LB containing 20 μg/mL cecB, and cells were grown at 37˚C for 24 hours. Error bars represent standard deviation (n = 3). **(A)** Representative growth curve of Δ*phoPQ*Δ*rcsB* and Δ*phoPQ*Δ*ompT* response to cecB. **(B)** Example of replicate in which Δ*phoPQ*Δ*rcsB* mutant failed to grow in the presence of cecB.
(TIF)

**S8 Fig. Inoculum effect in the Δ3 mutant.** MIC assays were conducted at the indicated dilution of an overnight culture of *E. cloacae* Δ3 mutant.
(TIF)

**S1 Table. Strains and plasmids used in this study.**
(DOCX)

**S2 Table. Oligonucleotides used in this study.**
(DOCX)

**S3 Table. Clinical isolates.**
(XLSX)

**S1 Data. Raw data used to generate figures.**
(XLSX)

## Acknowledgments

We thank Dr. Lars Westblade (Weill Cornell Medical) for *E. cloacae* MDR strains. We thank John Helmann and Heather Feaga for critical comments on the manuscript.

## Author Contributions

**Conceptualization:** Andrew N. Murtha, Misha I. Kazi, Tobias Dörr.

**Data curation:** Andrew N. Murtha, Misha I. Kazi, Eileen Y. Kim, Facundo V. Torres, Kelly M. Rosch.

**Formal analysis:** Andrew N. Murtha, Misha I. Kazi, Eileen Y. Kim, Facundo V. Torres, Kelly M. Rosch, Tobias Dörr.

**Funding acquisition:** Tobias Dörr.

**Investigation:** Andrew N. Murtha, Misha I. Kazi, Eileen Y. Kim, Facundo V. Torres, Kelly M. Rosch, Tobias Dörr.

**Methodology:** Andrew N. Murtha, Misha I. Kazi, Eileen Y. Kim, Facundo V. Torres, Kelly M. Rosch, Tobias Dörr.

**Project administration:** Tobias Dörr.

**Resources:** Tobias Dörr.

**Supervision:** Tobias Dörr.

**Validation:** Andrew N. Murtha, Misha I. Kazi, Eileen Y. Kim, Facundo V. Torres, Kelly M. Rosch, Tobias Dörr.

**Visualization:** Andrew N. Murtha, Misha I. Kazi, Eileen Y. Kim, Facundo V. Torres, Kelly M. Rosch, Tobias Dörr.

**Writing – original draft:** Andrew N. Murtha, Misha I. Kazi, Tobias Dörr.

**Writing – review & editing:** Misha I. Kazi, Tobias Dörr.

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
