## [Decision Letter · Decision Letter 0]

12 Jun 2024

Dear Dr. Dörr,

Thank you very much for submitting your manuscript "Multiple resistance factors collectively promote inoculum-dependent dynamic survival during antimicrobial peptide exposure in *Enterobacter cloacae*" for consideration at PLOS Pathogens. Again, we apologize for the delay during the review process. Your manuscript was reviewed by members of the editorial board and by three independent reviewers. The reviewers appreciated the attention to an important topic. Based on the reviews, we are likely to accept this manuscript for publication, providing that you modify the manuscript according to the review recommendations (below).

Please address all of the comments from the reviewers, especially the concerns of Reviewer 2 regarding the lack of an MDR strain and the use of cecropin B in your study.

Sincerely,

Daria Van Tyne

Academic Editor

PLOS Pathogens

D. Scott Samuels

Section Editor

PLOS Pathogens

Michael Malim

Editor-in-Chief

PLOS Pathogens

orcid.org/0000-0002-7699-2064

Reviewer Comments:

Reviewer's Responses to Questions

**Part I - Summary**

Reviewer #1: The manuscript by Murtha et al describes how Enterobacter populations resist killing by the antimicrobial peptide cecB. This study is important for numerous reasons. Antimicrobial peptides are a critical component of the innate immune response and resistance to or evasion of AMPs is necessary for to establish infection. Furthermore, AMPs are being explored as new treatment options, with some currently in clinical trials for the treatment of Gram-negative infection. Here, the authors characterize the response of Enterobacter to cecB and observe initial killing followed by delayed regrowth of the population. Interestingly, this phenomenon and the length of the lag phase following cecB treatment was heavily dependent on populations density and exhibited the hallmarks of heteroresistance. The authors then explored 3 hypotheses to uncover the mechanism of resistance, mutating genes responsible for three different established mechanisms of AMP resistance described in other organisms. Intriguingly, all three systems contributed to survival and regrowth of the population, likely due to heterogenous expression levels of each of the genes resulting in a sub-population exhibiting increased resistance to the AMP. Overall, this is a very well written, thorough and elegant paper that serves as a very useful addition to our knowledge of AMP resistance mechansims in an important pathogen.

Reviewer #2: The manuscript focuses on heteroresistance of E. cloacae to a cationic antimicrobial peptide (AMP) cecropin B. The authors demonstrate heteroresistance with growth curve experiments that show no heritable resistance occurs after challenge with cecropin B. This effect is inoculum dependent. The authors also confirm their data with CFU counts and quantitative microscopy. Then, the authors test 3 known systems in other bacteria that antagonize AMPs, PhoPQ, RcsB, and OmpT, in E. cloacae. The knockout of each of these systems results in reduced heteroresistance, allowing the authors to conclude that heteroresistance to cecropin B in E. cloacae is multifaceted. The work is done carefully with good controls, but I have two concerns about the impact/importance of the results. First, all experiments were done on the type strain of E. cloacae rather than MDR strains with more clinical/health relevance. Second, the authors do not motivate their choice of cecropin B sufficiently. Of all of the AMPs out there, why choose this one? It does not appear to be particularly potent, and it is a linear peptide which will impact its stability (indeed, likely observed in the authors’ OmpT experiments). Besides these overarching questions, there are several further questions/concerns that should be addressed in a revised paper.

1) Introduction: this section should include information about the sequence of cecropin B and its helical structure as well as if anything is known about whether it forms pores in membranes or kills by other mechanisms

2) Line 54: I am intrigued by the statement that AMPs kill faster than antibiotics. Could the authors discuss this more and back it up with numbers?

3) Line 86: while the intro is already on the long side, I would like the authors to add a few more sentence talking about the specifics of how CPS is regulated by Rcs. Up or down-regulated? Which polysaccharides?

4) Line 109: state the values of the “low MICs”

5) Line 114: report the concentration of cecropin B used

6) Line 128: I was confused about referring to the 1000-fold dilution as a standard condition because the paragraph only discusses the 10 and 100-fold dilutions

7) Line 184-185: amino arabinose is not an enzyme. Please correct and also state briefly how this enzyme leads to AMP resistance

8) Line 212: can you define the various components of the capsule, especially if there are aspects of it that are specific to E. cloacae?

9) Line 260: did the authors consider overexpression of the DH5a OmpT in E. cloacae to rule out the possibility that the lack of cleavage in DH5a is just due to lower expression?

10) Line 269: it should be possible to use LC-MS to try to quantify the degradation of cecropin B; was this considered?

Reviewer #3: Review: Murtha et al.

Overall, the experiments we well described, and the data clearly reported. The novelty is diminished by the target gene approach which only looked at known mechanisms, but it was still fruitful. The generalizability was diminished by the focus on just on AMP, and one bacterial strain. I believe there are a there are a few points of clarification, mostly on the interpretation that could make the paper stronger.

1. The justification of investigating this specific AMP is not made. Given the large number of AMPs that could have been chosen. It would have made a more compelling paper I the authors better explained why this AMP is of particular interest.

2. Their interpretation of the population level of heteroresistance assays in the results section is problematic. Specifically, the delayed growth of the population after initial decline. The authors do not address until much later, in the discussion, the clear alternative that the level of cecropin in the media is being depleted in a manner dependent upon the E. cloacae concentration. If this is the case, one could imagine there is no phenotypic heterogeneity among the E. cloacae within the culture, but simply a stochastic killing of cells until the cecropin level drops to a point where the cell can grow again.

-- Similarly, in the single cell microscopic studies, I don’t see that they fully explain why their findings may not be due to stochastic effects, and potentially local heterogeneity in cecropin concentration in the LB pads, which presumable could develop rather quickly if the E. cloacae are inactivating the cercopin.

-- The authors then lean heavily into the model into which cecropin is being depleted in the discussion ( eg line 348. ), which is also consistent with their data, but confusing given the inconsistency with how the data are interpreted in the results section.

-- Overall, I felt the frequent use of the term heteroresistance detracted from the paper, and I see no compelling evidence that heteroresistance (if defined as a phenotypic state the is vertically passed down for at least a few cell division cycles) exists.

3. The genetic studies including identification of the ompTecl with exploration to mutation in the active site was strong and well done.

**Part II – Major Issues: Key Experiments Required for Acceptance**

Reviewer #1: I have no major criticisms.

Reviewer #2: see above

Reviewer #3: None

**Part III – Minor Issues: Editorial and Data Presentation Modifications**

Reviewer #1: Can the authors more specifically introduce what is known about the mechanism by which cecropins kill. In the discussion, it may be worthwhile suggesting that the resistance mechanisms uncovered here may play more or less of a role in resistance to other AMPs depending on their mechanism of action.

The impact of population density on the outcome of AMP treatment is striking. Perhaps the authors would care to comment on the implications of this finding for the establishment of infection or for the use of AMPs to as new anti-infectives?

In Figure 7, 2. Membrane Stress/AMPs, consider removing AMPs as that’s already in the figure.

Reviewer #2: see above

Reviewer #3: none

Figure Files:

Data Requirements:

Reproducibility:

References:

---

## [Editor Report · Decision Letter 1]

8 Aug 2024

Dear Dr. Dörr,

We are pleased to inform you that your manuscript 'Multiple resistance factors collectively promote inoculum-dependent dynamic survival during antimicrobial peptide exposure in *Enterobacter cloacae*' has been provisionally accepted for publication in PLOS Pathogens.

Best regards,

Daria Van Tyne

Academic Editor

PLOS Pathogens

D. Scott Samuels

Section Editor

PLOS Pathogens

Michael Malim

Editor-in-Chief

PLOS Pathogens

orcid.org/0000-0002-7699-2064

---

## [Editor Report · Acceptance letter]

21 Aug 2024

Dear Dr. Dörr,

We are delighted to inform you that your manuscript, "Multiple resistance factors collectively promote inoculum-dependent dynamic survival during antimicrobial peptide exposure in *Enterobacter cloacae*," has been formally accepted for publication in PLOS Pathogens.

Best regards,

Michael Malim

Editor-in-Chief

PLOS Pathogens

orcid.org/0000-0002-7699-2064